# Experiences of Using WDumper to Create Topical Subsets from Wikidata

Seyed Amir Hosseini Beghaeiraveri[1][0000−0002−9123−5686] sh200@hw.ac.uk,
Alasdair J.G. Gray[2,3][0000−0002−5711−4872] a.j.g.gray@hw.ac.uk, and Fiona J.
McNeill[3][0000−0001−7873−5187] f.j.mcneill@ed.ac.uk

[1] Heriot-Watt University, Campus The Avenue, Edinburgh EH14 4AS, UK
[2] The University of Edinburgh, Edinburgh, UK

**Abstract.** Wikidata is a general-purpose knowledge graph covering a wide variety of topics with content being crowd-sourced through an open wiki. There are now over 90M interrelated data items in Wikidata which are accessible through a public query endpoint and data dumps. However, execution timeout limits and the size of data dumps make it difficult to use the data. The creation of arbitrary topical subsets of Wikidata, where only the relevant data is kept, would enable reuse of that data with the benefits of cost reduction, ease of access, and flexibility. In this paper, we provide a formal definition of topical subsets over the Wikidata Knowledge Graph and evaluate a third-party tool (WDumper) to extract these topical subsets from Wikidata.

**Keywords:** wikidata · knowledge graph subsetting · topical subset · wdumper.

## 1   Introduction

A Knowledge Graphs (KG) is defined as representing real-world realities as a graph, in which nodes are real-world entities and edges, are the relationships between them [10]. In recent years, there have been a growing number of publicly available KGs; ranging from focused topic specific ones such as GeoLinkedData [12], EventMedia [11], and UniProt [7], to more general knowledge ones such as Freebase [6], DBpedia [3], and Wikidata [16]. These general purpose KGs cover a variety of topics from sports to geography, literature to life science, with varying degrees of granularity.

Wikidata [16] is a collaborative and open Knowledge Graph (KG) created by the Wikimedia Foundation since 29 October 2012. The main purpose of Wikidata is to provide reliable structured data to feed other Wikimedia projects such as Wikipedia and is actively used in over 800 Wikimedia projects[3]. It contains over 90 million data items covering over 75 thousand topics[4]. Regular dumps of

---

[3] https://www.wikidata.org/wiki/Wikidata:Statistics accessed 4 February 2021

[4] https://w.wiki/yVY accessed 9 February 2021. Note that the query execution timesout if you try to return the count query `SELECT (COUNT(DISTINCT ?0) AS ?numTopics) WHERE { ?s wdt:P31 ?o }`.

the data are published under the Creative Commons CC0 public license. These dumps are available in several formats including JSON and RDF. However, the size of the gzipped download files has grown from  3GB in 2015 to 85GB, and keeps increasing as more data is added. The size of these files make it increasingly difficult and costly for others to download and reuse; particularly if only focused on a particular topic within the data, e.g. life sciences data or politicians. While Wikidata can be queried directly through an open SPARQL endpoint[5], it is subject to usage limits (as are other public endpoints of KGs) which limits the scale and complexity of queries that can be issued [16]. Thus there is a need for generating subsets over large KGs such that they contain the data for a specific topic in a size that enables complex analysis queries to be conducted in a cost effective and time efficient way.

Our motivation for research on Wikidata subsetting is a combination of research goals, flexibility and ease of use. From the flexibility and ease of use perspective, we are looking for Wikidata subsets that can allow users to run smaller versions of Wikidata on available platforms such as laptops and PCs. Wikidata, as a knowledge graph with an interesting data model, has significant features for inspiration and improvement, but the speed of research and the diversity of researchers will be reduced if any experiment on it requires powerful servers, processing clusters, and hard disk arrays.

From the research point of view, our motivation is creating a type of subsets we call *Topical Subset* which is a set of entities related to a particular topic and the relationships between them. Having topical subsets of Wikidata for example in the fields of art, life sciences, or sports, not only helps us achieve the first goal (flexibility and ease of use) but also provides a platform for comparing and evaluating Wikidata features in different topics. The subset will also make the research experiments reproducible as they can be archived and shared more easily.

You can envision that such subsets can be generated through SPARQL CONSTRUCT queries. While this is straightforward for small subsets focused on a single entity type, e.g. politicians, it does not scale to interrelated topics that make up a larger domain, e.g. the life sciences subset defined by GeneWiki [17].

In this paper, we present our experiences of defining and creating topical subsets over Wikidata using WDumper [1]; a third-party tool provided by Wikidata to create custom RDF dumps of Wikidata. We provide a formal definition for a KG subset on a specific topic (Section 3) and evaluate WDumper as a practical tool to extract such subsets (Section 4).

## 2   Knowledge Graph Subsets

General purpose KGs like Wikidata are valuable sources of facts on a wide variety of topics. On the Linked Data Web they serve as a common linking point

---

[5] https://query.wikidata.org/sparql accessed February 2021

between inter-, and sometimes intra-, domain KGs[6]. However, their increasing size makes them costly and slow to use locally. Additionally, the large volume of data in Wikidata increases the time required to run complex queries. This often restricts the types of queries that can be posed over the public endpoint since it has a strict 60-second limit on the execution time of queries. Any query that takes more time to execute than this will timeout[7].

Downloading and using a local version of Wikidata is a way of circumventing the timeout limit. However, this is not a cheap option due to the size of the data. A suggested system to have a personal copy of Wikidata includes 16 vCPUs, 128GB memory, and 800GB of raided SSD space[8]. A google cloud computation engine with these specification would cost more than \$570 per month[9].

There are a large number of use case scenarios where users will not need access to all topics in a large general purpose KG. A small and complete enough subset can be more suitable for many purposes. For example, a subset containing all data points about genes, proteins, drugs, and diseases would be useful in pharmaceutical research [17]. With a small subset inference strategies can be applied to the data and complete in reasonable time. Topical subsets could also be published along with papers, which provides better reproducibility of experiments [18]. Small subsets are also easier to archive. Various topical archives can be created from Wikidata, which gives better access to the data, while multiple time snapshots can be created from this data. Wikidata subsets can also provide more appropriate datasets for students and schools by censoring adult content. Therefore having a topical subset that is smaller but has the required data can enable complex query processing on cheap servers or personal computers — reducing the overall cost — whilst also providing an improvement in query execution times.

## 2.1   Topical Subset Use Cases

In this section, we define four examples of topical subsets in Wikidata. These examples will be our use cases to review WDumper and can also be used in other reviews as a comparison platform. Note that use cases are defined in terms of English language statements. A subsetting approach, method, or tool would need to formalise these, as appropriate for their configuration, to extract the relevant data.

*Politicians.* This subset should contain all entities that are an instance of the class *politician(Q82955)*, or any of its subclasses. The subset should contain all statements and properties pertaining to these entities.

---

[6] `https://lod-cloud.net/` accessed 9 February 2021

[7] `https://en.wikibooks.org/wiki/SPARQL/Wikidata_Query_Service/`

[8] See this post: `https://addshore.com/2019/10/your-own-wikidata-query-service-with-no-limits/`

[9] Estimated by Google Cloud Pricing Calculator: `https://cloud.google.com/products/calculator/#id=32eca290-7628-48af-9988-20508f4bc861` accessed 9 February 2021

This subset should contain all entities that are an instance of the class *politician*, or any of its subclasses. In the case of Wikidata, this would be the class (Q82955), while for DBpedia it would be the class (Politician). The subset should contain all facts pertaining to these entities, i.e. in Wikidata all statements and properties.

*General(military) Politicians.* The subset should contain all entities that are an instance of the class *politician(Q82955)* or any of its subclasses, who also are a *military officer(Q189290)* and have the rank of *general(Q83460)*, i.e. politico-military individuals.

The main goal of this use case is to see the effect of having more conditions in the English definition on the run-time and the volume of the output of subset extraction tools.

*UK Universities.* The subset should contain all instances of the class *university(Q3918)* or any of its subclasses, that also has *country(P17)* of the *United Kingdom(Q145)*. The subset should contain all statements and properties pertaining to these entities.

This use case extends the complexity of the subset by having alternative properties and values to satisfy.

*Gene Wiki.* This case is based on the class-level diagram of the Wikidata knowledge graph for biomedical entities mentioned in [17]. The goal of this paper and the Gene Wiki project is to make and maintain Wikidata as a central hub of linked knowledge on genes, proteins, diseases, drugs, and related concepts[10]. The class-level diagram specifies 17 different item types from Wikidata mentioned in the Gene Wiki project. The subset should contain all instances of the class *gene(Q7187)*, *protein(Q8054)*, and all other 17 items or their subclasses mentioned in the above class-level diagram.

The selection of these use cases is a combination of research and experimental goals. The GeneWiki and Politicians use cases have been selected for future research purposes because of their hypothetical richness in references. To have a quite smaller subset as output compared to the other two, we chose the UK universities. The General(military) Politicians use case was chosen because we want to see the effect of having more conditions in the subset creation compared to the Politicians.

## 2.2   Related Works

Creating a subset of Wikidata [2] was one of the topics covered in *12th International SWAT4HCLS conference*[11] and further pursued in project 35 of BioHackathon-Europe[12] [8]. Although there have been several proposals to do

---

[10] https://www.wikidata.org/wiki/Wikidata:WikiProject_Gene_Wiki

[11] http://www.swat4ls.org/workshops/edinburgh2019/

[12] https://github.com/elixir-europe/BioHackathon-projects-2020/tree/master/projects/35 accessed 9 February 2021

this, to the best of our knowledge there is no agreed definition for a topical subset nor a unified and evaluated way to create such subsets, especially a topical subset of Wikidata.

Matsumoto et al. [13] have introduced the Graph to Graph Mapping Language (G2GML) that aims to convert RDF graphs to property graphs. G2G Mapper[13] is a tool that receives a mapping config file written in G2GML and an RDF turtle file (or a SPARQL endpoint) as input and creates a property graph from the RDF data specified by the input mapping. Although the purpose of the G2GML language was to generate property graphs from RDF graphs to take advantage of the property graphs, it can be used as a topical subset creator, however the output will be a property graph.

Mimouni et al. [15,14] use a concept called the Context Graph to generate a smaller dataset than the original large KGs such as DBPedia and Wikidata which enable them to test their knowledge base completion method on this dataset instead of the entire KG. The context graph construction algorithm starts with an initial set of *seed nodes*, and in each round, adjacent nodes of the seed set (that are not in a forbidden set) and their relations are added to the seed nodes. This operation continues to a number of rounds called the *radius*. The context graph production process seems to be suitable for generating random subsets, however, it is not an integrated method for generating topical subsets. To produce topical subsets, as we can see in Section 3, we need a way to identify the member entities of a particular topic, but there is no such concept in the context graph. One has to extract all the nodes related to a topic from the beginning and put them in the initial seed set. On the other hand, extracting node neighbors to a $radius \geq 2$ may enter information that is not relevant to the topic.

WDumper [1] is a third-party tool for creating custom and partial RDF dumps of Wikidata suggested at the Wikidata database download page[14]. The WDumper backend uses the Wikidata Toolkit (WDTK) Java library to apply filters on the Wikidata entities and statements, based on a specified configuration that is created by its python frontend. This tool needs a complete JSON dump of Wikidata and creates an N-Triple file as output based on filters that the config file explains. This tool can be used as a topical subset creator, however, it cannot be said that WDumper can build a complete topical subset. This is due to the limitations of this tool, which we discuss in Section 4.5 after reviewing this tool.

Shape Expressions (ShEx) [9] is a structural schema language allowing validation, traversal and transformation of RDF graphs. The ability of keeping track of the triples employed during validation can be used to define data schemata which could help define subsets traversing the graph. Compared to WDumper, which is a tool focused on data extraction, ShEx is a language for validating RDF data, with the possibility of extracting traversed data with its developed tools. WDumper also works directly on the Wikidata JSON dump which making it the first choice for extracting topical subsets from Wikidata.

---

[13] Demo: `http://g2g.fun/`, Github: `https://github.com/g2glab/g2g`
[14] `https://www.wikidata.org/wiki/Wikidata:Database_download`

**Listing 1.1.** An example of a function $R$ which is a query to return all entities with type city

```
SELECT ?entity WHERE {
  ?entity wdt:P31 wd:Q515 . # instance of(P31) city(Q515)
}
```

## 3   Topical Subset Definition

We now provide a formal definition for topical subsets. This definition is based on Wikidata data model. Considering that Wikidata data model is RDF compatible, this definition can be generalized to all RDF-based KGs.

From the outside, the Wikidata knowledge graph consist of the following collections:

- $E$: set of Wikidata entities – their ID starts with a Q.
- $P$: set of Wikidata properties – their ID starts with a P.
- $S$: set of Wikidata statements.

Now we define the filter function $R : E \mapsto E$ as a black-box that can be applied on $E$ and selects a finite number of its members related to a specific topic. Let $E_R \subset E$ be the output of the function $R$. For entity $e \in E$ let $S_e \subset S$ be **all** simple and complex Wikidata statements in which $e$ is the subject. Note that in Wikidata, a simple statement is a regular RDF triple, while a complex statement is a triple that references and/or qualifiers attached to it. Also, let $P_e$ be **all** properties which are used in $S_e$ triples either for the statement itself or qualifiers/references. With these assumptions, we define dump $D_R$ as a topical subset of Wikidata with respect to R:

$$D_R := \{E_R, \bigcup_{e \in E_R} P_e, \bigcup_{e \in E_R} S_e\}$$

From the definitions of $P_e$ and $S_e$ we can conclude that $\bigcup_{e \in E_R} P_e \subset P$ and $\bigcup_{e \in E_R} S_e \subset S$ and subsequently D is a mathematical subset of Wikidata. We consider $R$ as black-box; the input of $R$ is the set of all Wikidata entities and its output is a subset of Wikidata entities related to a specific topic. The function $R$ can be any set of definitions, rules, or filters that describe a related group of entities. The definition of $R$ depends on the topic that is being described. One example of $R$ is a simple `SELECT` query that describes all entities that have type city (Listing 1.1).

## 4   WDumper

WDumper is a tool provided by Wikidata for producing custom dumps from Wikidata[15]. WDumper is capable of extracting some topical subsets as per our

---

[15] `https://www.wikidata.org/wiki/Wikidata:Database_download#Partial_RDF_dumps` - accessed 9 February 2021

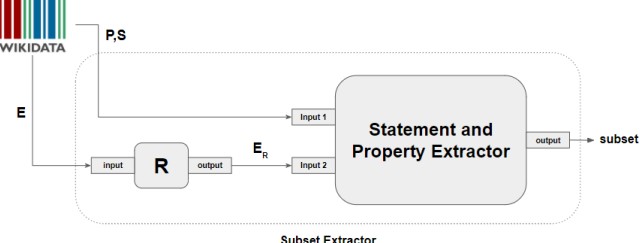

**Fig. 1.** Component overview of WDumper

definition. In WDumper, the $R$ function can be seen as a filtration approach on entities. For each topic, the appropriate filters on entities must be defined. Once the filters are defined and the subset $E_R$ is extracted, for each $e \in E_R$ WDumper extracts all statements with origin $e$ along with their qualifiers and references. A component overview of WDumper is given in Figure 1.

WDumper requires two inputs. The first input is the complete dump of the Wikidata database in JSON format. Note that Wikidata database is available with different formats such as JSON and N-Triple. We may refer to this complete dump as "full dump". The second input is a JSON specification file that contains rules and filters for determining which entities, properties and statements to extract from the full Wikidata dump.

The output of WDumper is an N-Triple (.nt) file that contains the entities and statements specified in the second input. There is also a GUI for creating the input specification file.

In this section we want to review WDumper . Our route for this review includes the following steps:

1. Generating WDumper specification files according to each use case of Section 2.1.
2. Running WDumper with the above specifications on two complete Wikidata dumps belonging to two different time points and compare the **run-time** and the **volume** of the extracted output.
3. Evaluating the extracted output via performing different queries both on the output and the input full dump.
4. Summarizing results and expressing strengths and weaknesses of WDumper.

### 4.1   WDumper Specification Files

In Section 2.1, we introduced some use cases that enable us to evaluate subset extraction tools. In this section, we describe how to generate the WDumper specification file according to each use case. The WDumper specification files for each use case can be found in [5].

*Politicians.* For achieving this use case with WDumper, we can define a filter on the *occupation(P106)* property of the entities to be a *politician(Q82955)*.

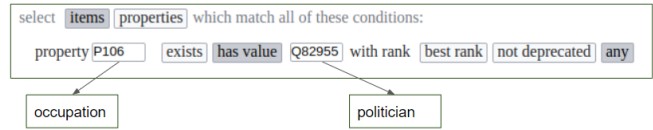

**Fig. 2.** Configuring WDumper GUI for the politicians subset.

Figure 2 Shows how to set the WDumper GUI for this example. Selecting the "any" option for the "rank" in Figure 2 forces WDumper to extract any entity that has an occupation property with the value politician even if this property is not the best rank or is deprecated. Note that in Wikidata, each statement can have a rank that can be used to identify the preferred value if the statement has more than one value. Ranks are available in Wikidata RDF data model like qualifiers and references.

*General(military) Politicians.* To achieve this subset, we add two more conditions into the filter of the above case. The first condition is *occupation(P106)* property to be *military officer(Q189290)*. The second condition is *military rank (P410)* property to be *general(Q83460)*.

*UK Universities.* For achieving this, we can add a filter on entities with two conditions: *instance of(P31)* property to be *university(Q189290)* and *country(P17)* property to be *United Kingdom(Q145)*.

*Gene Wiki.* For each item type of the class-diagram, we create a filter on entities in WDumper via *instance of(P31)* property to be *gene(Q7187), protein(Q8054)*, etc. In this case, different types are added as new basic filters, not new conditions as we want all different 17 types to be in the output by all their statements and properties.

Note that extracting properties is also possible like entities in WDumper (by selecting property instead of item in Figure 2). This feature as mentioned in the formal definition of topical subsets is required to extract Wikidata ontology on a particular topic. However in this paper we did not use this feature in our use cases to avoid extra complexity.

In addition to entities, WDumper can also apply filters on statements. For example, we can select whether specific statements (by mentioning their property) and their references and qualifiers be in the output or not. These filters, which are in the form of on-off buttons in the GUI, allow the WDumper to extract the intermediate nodes of statements, references, and qualifiers. In all use cases, we get a run without any filters on the statements and a run by selecting the option to include all references and qualifiers. The second run is to see the effect of statement filters on execution time and output volume. In the specification file, these filters can be seen in the statement sub-array, as "qualifiers", "references", "simple", "full" keys which are false or true respectively.

**Table 1.** Details of the Wikidata dumps. The size column states compressed JSON.gz size. Total items and total statements columns are obtained from Wikidata Stats tool[19] regarding to the specified dumps.

| Release Time | Size (GB) | Total Items | Total Statements |
|---|---|---|---|
| 2015-04-27 | 4.52 | 17,632,496 | 61,427,934 |
| 2020-11-13 | 90.42 | 90,368,519 | 1,153,949,899 |

### 4.2 Experimental Setup

We now give details of our experimental environment in which we will evaluate WDumper. This includes details of the Wikidata dumps and the hardware used.

*Input Dumps.* We use two full dumps of the Wikidata database. The first full dump is from 27 April 2015[16], and the second is from 13 November 2020[17]. The selected 2020 dump was the latest JSON dump available when conducting our evaluation. The selected 2015 dump is the first archive date for which both JSON and turtle files are available (we need the JSON file for WDumper running, while the turtle file is needed to import the full dump in a triplestore and evaluate output of WDumper based on the input). Table 1 provides summary information about these two dumps. The 2015 dump is smaller, can be stored and processed locally even on PCs, and it takes a much shorter time to generate output from it. For this reason, it is very suitable for initial tests. The 2020 dump, on the other hand, is much richer and can provide insights on how WDumper deals with large datasets of the size that Wikidata now produces.

*Experimental Environment.* All of WDumper running tests were performed on multi-core server with 64 8-core AMD 6380 2.5GHz 64bit CPUs, 512GB of memory and 2.7TB disk space. Java openJDK version 11 (build 11+28) and Gradle 6.7.1 was used to compile and run WDumper.

*Experimental Run.* The calculated times have been extracted from the elapsed time mentioned in WDumper output log. For each of the execution cases, three independent runs were performed and the average and deviation from the standard of these times were calculated.

### 4.3 Evaluating WDumper

We run WDumper with the specification files described in Section 4.1 and the two Wikidata dumps. The outputs can be found in [5]. The results of time and size

---

[16] Downloaded from `https://archive.org/download/wikidata-json-20150427` - accessed 11 November 2020

[17] Downloaded from `https://dumps.wikimedia.org/wikidatawiki/entities/` - accessed 15 November 2020

[19] `https://wikidata-todo.toolforge.org/stats.php`

are listed in Table 2. For each use case, we considered two types of specification files. First, a specification that aims to extract statement nodes, references, and qualifiers and is labeled with a "withRQFS" in Table 2. Second, a specification that aims to extract just simple statements without any statement nodes.

*Analysis.* The initial observations from Table 2 shows that the run-time on the 2020 dump is significantly longer than the 2015 dump. This can be justified by the larger volume of the 2020 dump and the increased data that must be processed to produce the subset. In all cases, the specification set with "withRQFS" took longer than the corresponding without "withRQFS", and produced more volume in the output, indicating the addition of references, qualifiers, and statements nodes in the output. For example, this change is very significant in the case of Gene Wiki in 2020 dump. The added filters as well as the conditions added to the filters also have a direct effect on the run-time and an inverse effect on the output volume, which is to be expected. Of course in run-times, the amount of data that must be written in the output must also be considered. This is evident, for example, in comparisons between UK universities and the military politicians in which the volume of data has a greater impact than the number of conditions. Overall, considering the high volume of data, the time required to extract a topical subset by WDumper seems appropriate. Also, adding more filters does not have a huge effect on runtime.

### 4.4   Topical Subset Validation

The previous section considered the runtime performance of the WDumper tool and the size of the generated subsets. We now consider validating the content of the subsets. That is, we consider if the produced output has the information that it was supposed to have according to the definition in Section 3. Our assessment will be based on the following hypotheses:

– **Hypothesis 1:** The number of filtered entities in the output should be equal to the same number of entities in the input dump. For example, in the Politicians use case, the number of persons with the occupation of politician in output should be equal to the number of persons with the occupation of politician in the corresponding input dump. This hypothesis can be tested with COUNT queries both on input and output.
– **Hypothesis 2:** For each entity that is supposed to be in output, the number of its related statements must be equal in both output and input dump. For example, in the Politicians use case, if the main dump has 50 statements about George Washington, we expect to see the same number of statements about this politician in the output too. This hypothesis can be tested using DESCRIBE queries.
– **Hypothesis 3:** WDumper can extract intermediate statement nodes, references, and qualifiers exactly as they are at the input dump. This hypothesis can be tested by querying the qualifiers and references of some given statements.

**Table 2.** Time elapsed and the size of the output in running WDumper on each use case and full dumps. "withRQFS" label denotes the specification that aims to extract statement nodes, references, and qualifiers. The FoE column denotes the number of filters on entities. The CiF column denotes the number of conditions in each filter. The times are averaged from three runs (AVG column) and the standard deviation (SD column) is also written. Sizes belong to compressed nt.gz files that are the direct output of WDumper. Inside parentheses, sizes and times are converted to other units for more readability (h for hours, KB for kilobytes, and MB for megabytes).

| Use case | FoE | CiF | 2015 Dump | | | 2020 Dump | | |
|---|---|---|---|---|---|---|---|---|
| | | | Time (sec) | | Size (GB) | Time (sec) | | Size (GB) |
| | | | AVG | SD | | AVG | SD | |
| Politicians | 1 | 1 | 1835 (31min) | 198 | 0.07 (70MB) | 39779 (11h) | 2029 | 0.37 (370MB) |
| Politicians withRQFS | 1 | 1 | 2347 (39min) | 28 | 0.31 (300MB) | 44271 (12h) | 2811 | 1.4 |
| UK Universities | 1 | 2 | 1584 (26min) | 38 | 0.000105 (105KB) | 41474 (12h) | 3424 | 0.000255 (255KB) |
| UK Universities withRQFS | 1 | 2 | 1774 (30min) | 142 | 0.000175 (175KB) | 41890 (12h) | 8436 | 0.000864 (864KB) |
| General (military) Politicians | 1 | 3 | 1570 (26min) | 59 | 0.000268 (268KB) | 37155 (10h) | 917 | 0.00105 (1MB) |
| General (military) Politicians withRQFS | 1 | 3 | 1655 (28min) | 48 | 0.000664 (664KB) | 42334 (12h) | 7341 | 0.04 (4MB) |
| Gene Wiki | 17 | 1 | 1731 (29min) | 95 | 0.01 (11MB) | 40828 (11h) | 2284 | 0.70 (709MB) |
| Gene Wiki withRQFS | 17 | 1 | 1844 (31min) | 276 | 0.026 (26MB) | 48943 (14h) | 4993 | 4.5 |

*Evaluation Environment.* The environment used to evaluate is the same as the running experiment environment in Section 4.2. Testing the hypotheses requires running different queries on input and output of WDumper. For the output of WDumper, we use Apache Jena triple store version 3.17 to import data as TDB2 RDF datasets and perform queries.

*Data Corrections.* For the evaluation process, we require to perform some SPARQL queries over the input and the output of the WDumper. We tried to import the turtle version of 2015 dump [20] in Jena, which has a smaller volume but we recognized that the available turtle file has some syntax problems. One of these syntax problems, which can be seen in more than 100 cases in 2015 dump, is the bad end lines in the file as we can see in Figure 3. Another type of error that was observed is the existence of characters such as '\a' that cannot be read by Jena. Unacceptable characters such as '\n' and '\\' can also be seen in the

---

[20] Downloaded from `https://archive.org/download/wikidata-json-20150427` - accessed 20 December 2020

**Fig. 3.** An example of bad end lines in the Wikidata 2015 turtle dump file.

**Listing 1.2.** Commands used for fixing syntax errors of the 2015 dump.

```
sed -i -E 's/(<.*)}(.*>)/\1\2/' <dump_file>
sed -i -E 's/(<.*)\\n(.*>)/\1\2/' <dump_file>
sed -i -E 's/(<.*)\|(.*>)/\1\2/' <dump_file>
```

WDumper outputs, which reinforces the possibility that this problem occurs due to the conversion of information from the JSON file to RDF format.

In the case of WDumper outputs and the 2015 dump, we fixed the errors manually by a set of *sed* commands in Listing 1.2. The sanitized dump is available in [4]. We then imported the WDumper outputs and the 2015 dump into Apache Jena. In the case of 2020 dump, we use Wikidata Query Service (WDQS) because importing the full 2020 Wikidita dump data in the turtle (.ttl) format with Apache Jena with a size of 150GB requires days of processing. We started the importing process but even the 2020 dump has the syntax errors which require much more time to correct. The date of implementation of our evaluation queries is approximately two months after the creation date of the 2020 dump (November, 27). In this period, new data may have entered Wikidata which are available by WDQS and are not present in 2020 dump (and subsequently are not present in the WDumper output). Because of this, there may be slight differences in the counts of entities and statements between input and output that is not related to WDumper functionality. We tried to use Wikidata history query service[21] to quantify the rate of Wikidata increases in this period, but the history covers a range from the creation of Wikidata to July 1st 2019.

**Validation of Hypothesis 1** We use COUNT queries to validate this hypothesis. The purpose of these queries is to count the entities that should be in the output according to the filter(s) of each use case. If WDumper is performing correctly, the result of this count should be the same on both the output subset and the input dump. These queries will be different for each use case, depending on the definition of that use case. For example, while in the Politicians use case we count the number of people with political jobs, in the case of Gene Wiki we count the union of entities of type disease, genes, proteins, etc.

Listing 1.3 shows the queries executed for each use case. These queries run on the each use case's output ("withRQFS" version only) and the corresponding input dump separately. In Table 3 the results of performing COUNT queries are shown.

---

[21] https://www.wikidata.org/wiki/Wikidata:History_Query_Service

**Listing 1.3.** COUNT queries for evaluating hypothesis 1. Prefixes and most of Gene Wiki's query have been deleted for more readability.

```
############ Politicians ##############################
SELECT (COUNT (DISTINCT ?item) AS ?count) WHERE{
?item wdt:P106 wd:Q82955 . # occupation of politician
}

############ UK Universities ##########################
SELECT (COUNT (DISTINCT ?item) AS ?count) WHERE{
?item wdt:P31 wd:Q3918 ; # instance of university
      wdt:P17 wd:Q145  . # country of United Kingdom
}

############ General(military) Politicians ############
SELECT (COUNT (DISTINCT ?item) AS ?count) WHERE{
?item wdt:P106 wd:Q82955  ; # occupation of politician
      wdt:P106 wd:Q189290 ; # occupation of milit. officer
      wdt:P410 wd:Q83460  . # military rank of general
}

############ Gene Wiki ##############################
SELECT (COUNT (DISTINCT ?item) AS ?count) WHERE{
{?item wdt:P31 wd:Q423026 .}   # instance of active site
UNION
{?item wdt:P31 wd:Q4936952 .}  # instance of anat. struct.
UNION
# ...
UNION
{?item wdt:P31 wd:Q50379781 .} # instance of therap. use
}
```

**Table 3.** Results of performing COUNT queries of each use case (Listing 1.3) on the output of WDumper and input full dump for both 2015 and 2020 dumps. The last column are COUNT results queried against WDQS instead of the 2020 dump itself.

| Use case | 2015 Dump | | 2020 Dump | |
|---|---|---|---|---|
| | Output | Input | Output | Input |
| Politicians | 246,009 | 246,044 | 641,387 | 646,401 |
| General (military) Politicians | 165 | 165 | 597 | 602 |
| UK Universities | 73 | 73 | 183 | 186 |
| Gene Wiki | 19,432 | 19,432 | 3,282,560 | 3,283,471 |

*Analysis.* Table 3 shows that for 2015 dump, the number of entities in the output and input is equal except for the Politicians use case. In both 2015 and 2020 dumps, the difference between input and output is less than one percent in the cases of inequality. In the case of 2020 dump, the difference can be attributed to the entry of new data in the interval between our tests and the dump date. This is reasonable especially in the case of Gene wiki which many bots are importing new information into Wikidata every day. In the case of 2015 dump in the Politicians row, the 35 differences between input and output is unjustifiable. The reason for this difference may be the inability of WDumper to parse the data of these entities in the input dump. WDumper uses the JSON file as input, and to be able to fetch an entity, it must see the specific structure of the Wikidata arrays and sub-arrays in the JSON file. Some entities may not have this complete structure in the JSON file but they do exist in the turtle file.

**Validation of Hypothesis 2** To validate this hypothesis, in each use case we use DESCRIBE queries for an arbitrary entity that is in the WDumper output. The purpose of DESCRIBE query is to list all triples of the given entity. We

**Table 4.** Results of performing DESCRIBE queries on the selected entity output of WDumper and input full dump for both 2015 and 2020 dumps in each use case.

| Use case | Entity | 2015 Dump | | 2020 Dump | |
|---|---|---|---|---|---|
| | | Output | Input | Output | Input |
| Politicians | Q23 | 408 | 776 | 871 | 921 |
| General (military) Politicians | Q355643 | 104 | 150 | 207 | 228 |
| UK Universities | Q1094046 | 64 | 108 | 208 | 224 |
| Gene Wiki | Q30555 | 12 | 22 | 30 | 37 |

**Table 5.** Details and total numbers of predicates that are in the 2015 dump and WDumper could not fetch.

| Entity | schema:name | skos:prefLabel | Total |
|---|---|---|---|
| Q23 | 184 | 184 | 368 |
| Q355643 | 23 | 23 | 46 |
| Q1094046 | 22 | 22 | 44 |
| Q30555 | 5 | 5 | 10 |

expect that the result of DESCRIBE should be the same on both the output subset and the input dump. For each use case, we selected an arbitrary entity (called Tested Entity) which is present both in the input dump and the output of WDumper. Then we run a "DESCRIBE wd:Q..." query and count the extracted triples. Table 4 shows the results of performing describe queries on both input dump and output of WDumper in both 2015 and 2020 cases.

*Analysis.* From Table 4 it is clear that the number of triples in the DESCRIBE queries in both 2020 and 2015 dumps are not equal. This difference prompted us to explore the differences using the compare module of the RDFlib library. It was found that in the case of the 2015 dump, the input dump contains predicates such as `<http://www.w3.org/2004/02/skos/core#prefLabel>` and `<http://schema.org/name>`, which are not extracted by WDumper. Table 5, Shows the details and total numbers of predicates that are in the input dump (2015 dump) for the selected entities and WDumper could not fetch. As we can see, the total column is exactly the difference between the output and the input in the 2015 dump at Table 4. In the case of 2020 dump, some of predicates with `<http://www.w3.org/2004/02/skos/core#>` prefix, such as `dateModified`, and all `<http://www.wikidata.org/prop/direct-normalized/>` predicates are not detectable by WDumper. However, in both dumps the statements whose predicate is a property of Wikidata (e.g. P31, P106, etc.), completely extracted by WDumper.

**Validation of Hypothesis 3** To validate this hypothesis, we selected an arbitrary entity from each use case, and for this entity, we considered one of its statements. We then counted the qualifiers and the references of this statement in 2020 dump (over the WDQS) and in the output of WDumper. Table 6 shows

**Table 6.** Number of qualifiers and references for the selected property of the selected entity in the output and input of WDumper (2020 dump).

| Entity | Property | Qualifiers | | References | |
|---|---|---|---|---|---|
| | | Output | Input | Output | Input |
| Q23 | P26 | 4 | 4 | 2 | 2 |
| Q355643 | P485 | 1 | 1 | 1 | 1 |
| Q1094046 | P355 | 1 | 1 | 1 | 1 |
| Q17487737 | P680 | 24 | 24 | 96 | 96 |

the selected entity, selected property, and the number of qualifiers and references for them. From Table 6, it is clear that WDumper can extract qualifiers and references completely from the input.

### 4.5   Summarizing Results, Strengths and Weaknesses

The results of our evaluations show that WDumper, as a custom dump production tool, can be used to create some topical subsets. This tool can correctly and completely extract the entities specified by its filters. It also extracts almost all statements related to entities (except it is not designed to extract some prefixes). One of the features we have been looking for is the ability to extract references and qualifiers of Wikidata statements, which WDumper can do. Setting up this tool is not very complicated; the user only needs to select the filters of the entities and statements, run the tool and it extracts all of the information at once. Its GUI is also somewhat helpful, while the JSON structure of its specification files is also simple and understandable.

**Limitations and Weaknesses** WDumper tool has some weaknesses that we address in this section. The most important weakness of WDumper with regard to the topical subsets is the limitation in the definition of entity filters. In WDumper, entities can only be filtered based on the presence of a $P_x$ property or having the value $v$ for a $P_x$ property. Although it is possible to deploy any number of such filters, this is not enough to specify some kinds of use cases. For example, suppose we want to specify the Scottish universities subset. By reviewing some of these universities on the Wikidata website, we can find that their corresponding entity does not have any property that directly indicates they belong to Scotland. Of course, we can define the $R$ function of these subsets through indirect methods (for example, considering the Geo-location of entities in Scottish universities), but these type of filters are not available in WDumper.

The recognition of type hierarchies is another limitation of WDumper. In the case of UK universities, for example, the *University of Edinburgh(Q160302)* is not among the universities extracted by WDumper. The reason for this is that property *instance of(P31)* in this university refers to *public university(Q875538)* instead of *university(Q3918)*. In SPARQL queries, such cases are handled by the property paths like `wdt:P31/wdt:P279`*. These property paths are not available

in WDumper and considering more filters for each subtype needs to be familiar with the Wikidata ontology and will fail if the class hierarchy changes.

Another limitation is the ability to communicate between different filters in multi-filter cases. For example, in the Gene Wiki use case, we may want diseases that are somehow related to a gene or protein, while in WDumper output there are diseases that have nothing to do with genes, proteins, and other Gene Wiki types. The ability to choose another output format other than N-Turtle, especially the Wikibase JSON output, which is more suitable for using the subset produced in a Wikibase instance and also has a smaller volume, is another limitation.

The main implications of these limitations is the reduction of flexibility of subset extracting with this tool. With these weaknesses, users have to spend much more time defining the desired subset.

## 5    Conclusion

In this paper, we reviewed the issue of building topical subsets over the Wikidata knowledge graph. Our motivation for topical subsets is to enable efficient evaluation of complex queries over the knowledge graph with lower costs, reproducibility of experiments through archiving datasets, ease of use, and flexibility. We provided example use cases for topical subsets as well as a formal definition for topical subsets. This definition enables us to evaluate and compare subset creation tools.

In this study we used WDumper for topical subset extraction over Wikidata and then tested it by measuring run-time and output volume on four different use cases. We evaluated the correctness of the subsets generated by WDumper by comparing the answers to queries over the subsets and the full knowledge graph. Our experience shows that WDumper **can be used** to generate topical subsets of Wikidata in some use cases but **not all use cases**. WDumper can extract the entities specified by its filters and extract most statements related to those entities; it also fetches the statement nodes and references/qualifiers. However, WDumper has some weaknesses regarding to topical subsets. Its main problem is the way it defines filters on entities that reduces the power of this tool to build topical subsets. The most tangible issue is the **inability to define and fetch subclasses** of a class of entities, which is important in many use cases. Our suggestion for the future works is to explore alternative subsetting approaches such as using SPARQL queries or Shape Expressions. With selectors like SPARQL queries or ShEx schemata, we can increase the expressivity of the subset creation. It will also allow for subsets to be created on Knowledge Graphs other than Wikidata.

Acknowledgement. We would like to acknowledge the fruitful discussions with the participants of project 35 of the BioHackathon-Europe 2020; Dan Brickley, Jose Emilio Labra Gayo, Eric Prud'hommaux, Andra Waagmeester, Harold Solbrig, Ammar Ammar, and Guillermo Benjaminsen.

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
