# OpenReview forum: "Experiences of Using WDumper to Create Topical Subsets from Wikidata"
_eswc-conferences.org/ESWC/2021/Workshop/KGCW — KGCW 2021_

### Official Review · ~Dylan_Van_Assche1 · 2021-04-13
**Interesting and clear evaluation of the WDumper tool to create topical subsets of knowledge graphs.**

**Rating:** 7
**Confidence:** 4

**Review:**

This paper presents the experiences of using the WDumper tool for creating subsets of the Wikidata knowledge graph.
This tool is used to address the following problems: the increasing size of Wikidata data dumps makes it hard to process these dumps (scalability)
or to answer questions using these data (limited query complexity) if there is only an interest in data about a specific topic.

The paper evaluates the WDumper tool using four example use cases: Politicians, General (military) Politicians, UK Universities, Gene Wiki. A list of strengths and weaknesses are listed of WDumper for creating topical subsets.

## Section 2

In Section 2.1 several example use cases are proposed which are used to evaluate WDumper to create topical subsets. The goal of each use case is clearly explained. My main questions are:
- *How did the authors select these use cases?*
- *Are there no existing use cases available from related work?*

These questions were mostly answered in Section 2.2 where related work is discussed.
From the reader's point of view, it is interesting to discuss the related work first, this makes it more clear why the authors made certain decisions about the use cases they propose to evaluate the WDumper tool.

## Section 3

Section 3 presents a definition for topical subsets of RDF-based Knowledge Graphs.
Since related work doesn't provide such a definition, the authors defined 'topical subsets' based on the Wikidata data model, which is RDF compatible.

*I was wondering if this definition could be easily expanded to graphs in general, instead of only RDF-based Knowledge Graphs?*

## Section 4

Section 4 evaluates the WDumper tool on 2 Wikidata dumps and clearly summarizes the results, strengths and weaknesses of the WDumper tool.
In the introduction of this section, the paper claims that the WDumper tool is capable of extracting **some topical subsets**. It would be useful to provide a couple of examples here for the reader which topical subsets are possible and which are not.

During the evaluation of the WDumper tool, 2 Wikidata dumps are used.
This evaluation provides several insights into the feasibility of using the WDumper tool as a topical subset generator. The selection of both Wikidata dumps are interesting as they show how Wikidata has grown over the last 5 years. *The paper could benefit from adding more Wikidata data dumps to show the evolution of the volume compared to the run-time of the WDumper tool, for example: dumps of 2016, 2017, 2018, and 2019.*

## Typos, misc.:

*In order of appearance:*

- google cloud → Google Cloud
- python → Python
- that have type city (Listing 1.1) → that have type 'city' (Listing 1.1)
- In this section we want to review WDumper . → unnecessary space before '.'
- This feature as mentioned in the formal definition of topical subsets is required to extract → commas are missing
- their references and qualifiers be in the output or not → 'to' is missing before 'be'
- turtle → Turtle
- Of course in run-times, the amount of data → shouldn't 'in' be 'besides'?

---

### Official Review · ~Aidan_Hogan1 · 2021-04-14
**Interesting problem but limited technical contribution**

**Rating:** 6
**Confidence:** 4

**Review:**

The paper discusses a practical (and I would assume, very common) problem faced by people that wish to work with a topical subset of a large knowledge graph. Specifically, if the topical subset is too large, then one cannot use the query service as the query will time out. The other option is to download a dump and extract the subset locally, but this leaves the problem of how to define the subset and extract it. In this context, the authors evaluate a third-party tool, called WDumper, which is provided by Wikidata. The tool takes as input the full Wikidata JSON dump and allows for selecting a subset of entities from Wikidata based on having selected property-object pairs; all statements with a selected subject are then output in RDF format. The authors define a number of topical subsets, and run some experiments to test the performance and correctness of WDumper for a 2015 and a 2020 dump of Wikidata. They also discuss some of the limitations of the tool, such as the inability to select entities based on more complex criteria (most notably, based on being an instance of some super-class).

In terms of the strengths of the paper:

S1: The problem identified is probably a significant “barrier-to-entry” for potential adopters of large knowledge graphs, such as Wikidata, and one that is only going to grow worse as Wikidata continues to grow, and thus I agree that it is important to address. The problem is also non-trivial in the sense that there is an interesting trade-off between expressivity in terms of providing detailed criteria for defining the topical subgraph, and scalability/efficiency in terms of being able to extract that subgraph from the full dump. At one end of the spectrum, we have SPARQL, which allows for specifying detailed criteria, but is lacking in terms of scalability/efficiency when dealing with large inputs and outputs; at the other end of the spectrum, we have WDumper, which only supports simple criteria, but can compute the subgraph by means of a single scan of the dump (or at least it should be possible to process these criteria in a streaming way, document-by-document).

S2: Aside from discussion, there are experiments that provide concrete data and insights into the operation of the WDumper tool, in terms of efficiency and correctness.

S3: I found the discussion of the limitations of WDumper to be perhaps the most interesting part of the paper, as it suggests some concrete open challenges towards solving the problem. (I would add the limitation that one still needs to download and process a massive dump, even if they are only interested in a fraction of the data!)

S4: There is a comprehensive discussion of related works.

In terms of the weaknesses of the paper:

W1: The authors do not really propose something new here, but rather analyse/evaluate an existing tool. The analysis of the tool itself is not particularly deep, rather presenting some runtimes and correctness checks (though I can appreciate that when working with large datasets and long-running processes, considerable time and effort is required). Overall I think that the technical contribution is minimal, even considering that this is a workshop paper.

W2: The paper gives me the impression at times of trying to superficially "tick boxes" in terms of what a research paper is expected to have. There are two instances of this: quoting "hypotheses" that are not scientific hypotheses, but rather basic conditions for correctness, and providing formal definitions that are buggy and (in my opinion) unnecessary. Both the hypotheses and formal definitions feel forced to me.

More specifically, I found the use of the term "Hypothesis" in Section 4 to be strange. I guess that it might be okay in a very literal sense ("we hypothesise that the output is correct in terms of X"), but it seems to be atypical usage (i.e., not related to a scientific hypothesis). What the authors describe relates more to a basic test of the correctness of the output. I would rather call these by a title like "Condition".

Regarding the formal definitions, Section 3 provides a formal treatment of what is a "Topical Subset Definition". What is described here is a filter function that selects a subset of entities from Wikidata, and a topical subset extraction that outputs all statements whose subjects have been selected by the filter. While it is a commendable aim to be precise, my concern is that: (a) the notation is buggy and imprecise; (b) the authors do not provide a general definition of a topical subset, but rather of what WDumper supports in terms of topical subsets, where the definitions thus lack generality (e.g., can we not have topic subsets that include statements where a selected entity appears as an object?); (c) what is defined is trivial and does not require formal definition. Specifically regarding the notation being buggy:

- "$R : E \mapsto E$" This notation is a bit broken. Perhaps the authors mean to say: "$R : 2^E \rightarrow 2^E$" ($R$ maps a set of entities to a set of entities). The $\mapsto$ symbol suggests that $R(E) = E$, i.e., it returns the set of nodes itself.
- In the definition of $D_R$, better to use a tuple $( \ldots )$ rather than a set $\\{ \ldots \\}$ as the three elements are not alike.
- "and subsequently D" -> "and subsequently $R_D$"?
- "is a mathematical subset of Wikidata" The wording is a bit strange. It suggests that there are non-mathematical subsets that the authors wish to distinguish their definitions from.
- The definitions are very incomplete in the sense that a statement is never defined, a qualifier is never defined, etc., so the section mixes formal and informal definitions in a seemingly arbitrary way, and in a way that I fear ends up suffering the disadvantages of both forms of presentation, being both imprecise and unintuitive at the same time.


Overall, I think that the paper presents a limited technical contribution, even for a workshop paper, and have concerns about how parts of the paper are written. However, I think the problem of selecting topical subsets of a large knowledge graph is interesting, that this workshop would be a good place to discuss this problem, and that the paper could encourage such a discussion. Thus, overall, I lean slightly towards an accept.

If accepted, I would encourage the authors to rethink how they present Sections 3 and 4. I would maybe even suggest to remove Section 3 entirely and replace it with a brief informal definition. In Section 4, I would not talk about "hypotheses" but rather "conditions" or "tests". I would also suggest to review the following minor comments.


 Minor comments:

- The workshop calls for papers of 12-15 pages. The paper is 16 pages without references and 17 pages with references.
- "A Knowledge Graphs (KG)" -> "A Knowledge Graph (KG)"
- "a type of subset[] we call Topical Subset"
- "To have a [considerably] smaller subset"
- In terms of related works, YAGO4 has also been defined as a selected subset of Wikidata based on schema.org. Though it does not allow for selecting topics, the use of notability is an interesting way to reduce the dataset size, and I think it fits within the broader scope of approaches for selecting more manageable subsets of Wikidata (so probably worth including in the discussion on related works).
	Thomas Pellissier Tanon, Gerhard Weikum, Fabian M. Suchanek:
	YAGO 4: A Reason-able Knowledge Base. ESWC 2020: 583-596
- "the Wikidata knowledge graph consist[s]"
- "In this section we want to review WDumper[.]"
- "the case of Gene Wiki in [the] 2020 dump"
- "queries on [the] input and output of WDumper"
- "[were] completely extracted by"
- "Another limitation is the [in]ability"
- "while in [the] WDumper output"
- "has some weaknesses regarding [] topical subsets."

---

### Official Review · ~Umutcan_Simsek1 · 2021-04-18
**experiment with a useful tool with some significant issues**

**Rating:** 6
**Confidence:** 3

**Review:**


the paper presents an evaluation of the WDumper tool provided by Wikidata, in order to get topical subsets of Wikidata.

The paper motivates need for such a tool very well. It also provides an evaluation (which I find partially not very sound) and lists the limitations. Related work is also discussed in good detail. It would have been beneficial however if the technical details of the tool were also presented (how the JSON dumps are queried in the memory, is there any indexing etc.). I think presenting an experience with such a tool may be beneficial for the workshop, but I see some very major improvements (especially the last point below) to be made.

here are the issues with the paper from my point of view:

- beginning of section 2 is a bit repetition of the introduction.

- there are several use cases considered. I do not understand how the UK universities use case provides alternative properties to satisfy? I do not see any disjunctive query in the use case definition. Is it meant to provide alternative properties comparing to other use cases?
- the Gene Wiki use case mentions 17 different item types. do they include gene and protein types?
- In the related work, there is a sentence like "WDumper also works directly on the Wikidata JSON dump which making
it the first choice for extracting topical subsets from Wikidata." This sentence would indicate that ShEx also works on JSON dumps but it is not clear if this is the case and why would that make WDumper the first choice.
- why does it matter to make a distinction between qualifier statement and a regular RDF triple. RDF model of Wikidata uses a form of reification (n-ary relationship pattern) so everything can be represented with triples. Without formal definitions of these it is hard to say, and it appears the distinction has no signifcant impact on how the tool works. Isn't every RDF triple a regular RDF triple? RQFS statements are just triples with special predicates (they obviously increase the number of RDF triples to be considered)
- the hypotheses actually boil down to completeness of the tool results, which is rather a typical evaluation dimensions for such a task. than three hypotheses.
- I think the most significant shortcoming of the paper is the usage of DESCRIBE queries in the evaluation. Relying on describe queries are tricky, as they are implementation-dependent. for example I made a quick test with Wikidata Query Service with the entity Q23 [1]. The query returns over 2000 statements as it also includes the statements where Q23 is an object. The way DESCRIBE queries are implemented may be different with the triple store Wikidata uses and Jena TDB/SDB. Instead of the formalizations that are not really used again in the paper, more query examples can be provided for the evaluation. As it is currently not clear if the DESCRIBE queries were further conditioned on the item being the subject of the statements.

[1] https://query.wikidata.org/#DESCRIBE%20wd%3AQ23

---

### Meta-Review · Program_Chairs · 2021-04-21

**Recommendation:** Accept
**Confidence:** 5

**Metareview:**

The three reviewers mentioned several improvements that have to be considered for the camera-ready version:
- Review formal definitions and fix them
- Hypotheses are not actual research hypotheses, consider changing the term used to define that "conditions"
- Give more value to the research problem you are tackling, as it is really relevant for the workshop topics
- Typos and minor comments should be also considered.

David

---

### Decision · Program_Chairs · 2021-04-23

Accept